# The Effect of Pressotherapy on Performance and Recovery in the Management of Delayed Onset Muscle Soreness: A Systematic Review and Meta-Analysis

**DOI:** 10.3390/jcm11082077

**Published:** 2022-04-07

**Authors:** Paweł Wiśniowski, Maciej Cieśliński, Martyna Jarocka, Przemysław Seweryn Kasiak, Bartłomiej Makaruk, Wojciech Pawliczek, Szczepan Wiecha

**Affiliations:** 1Department of Physical Education and Health in Biala Podlaska, Faculty in Biala Podlaska, Jozef Pilsudski University of Physical Education in Warsaw, 21-500 Biala Podlaska, Poland; pawel.wisniowski@awf.edu.pl (P.W.); maciej.cieslinski@awf.edu.pl (M.C.); martyna.jarocka@awf.edu.pl (M.J.); bartlomiej.makaruk@awf.edu.pl (B.M.); wojciech.pawliczek@awf.edu.pl (W.P.); 2Students’ Scientific Group of Lifestyle Medicine, 3rd Department of Internal Medicine Cardiology, Medical University of Warsaw, 02-091 Warsaw, Poland; przemyslaw.kasiak1@gmail.com

**Keywords:** pressotherapy, compression, regeneration, DOMS

## Abstract

Background: It has been demonstrated that pressotherapy used post-exercise (Po-E) can influence training performance, recovery, and physiological properties. This study examined the effectiveness of pressotherapy on the following parameters. Methods: The systematic review and meta-analysis were performed according to PRISMA guidelines. A literature search of MEDLINE, PubMed, EBSCO, Web of Science, SPORTDiscus, and ClinicalTrials has been completed up to March 2021. Inclusion criteria were: randomized control trials (RCTs) or cross-over studies, mean participant age between 18 and 65 years, ≥1 exercise mechanical pressotherapy intervention. The risk of bias was assessed by the Cochrane risk-of-bias tool for RCT (RoB 2.0). Results: 12 studies comprised of 322 participants were selected. The mean sample size was *n* = 25. Pressotherapy significantly reduced muscle soreness (Standard Mean Difference; SMD = −0.33; CI = −0.49, −0.18; *p* < 0.0001; I^2^ = 7%). Pressotherapy did not significantly affect jump height (SMD = −0.04; CI = −0.36, −0.29; *p* = 0.82). Pressotherapy did not significantly affect creatine kinase level 24–96 h after DOMS induction (SMD = 0.41; CI = −0.07, 0.89; *p* = 0.09; I^2^ = 63%). Conclusions: Only moderate benefits of using pressotherapy as a recovery intervention were observed (mostly for reduced muscle soreness), although, pressotherapy did not significantly influence exercise performance. Results differed between the type of exercise, study population, and applied treatment protocol. Pressotherapy should only be incorporated as an additional component of a more comprehensive recovery strategy. Study PROSPERO registration number—CRD42020189382.

## 1. Introduction

Physical activity, especially at the competitive level, causes a lot of negative changes in the human body [1,2]. Inflammation occurs as a result of damage to muscle cells [3] from which creatine kinase (CK), lactate dehydrogenase, and metabolites are released [1,2]. In such cases, we observe decreased efficiency, faster muscle fatigue, a decrease in the range of motion (ROM), and the appearance of pain in places where they are overloaded [4,5]. This phenomenon is exacerbated especially with eccentric exercises (ECC) [6], in which intense exercise may cause Delayed Onset Muscle Soreness (DOMS) [7].

To increase exercise capacity as well as reduce the risk of injury, the key element is the use of training measures related to biological recovery to reduce metabolites to minimum values and to ensure the right amount of energy substrates, including ATP and phosphocreatine [8].

The most commonly used methods of biological recovery include treatments in the field of physical therapy (cold therapy, heat therapy, electrotherapy, compression therapy), manual therapy, massage (myofascial release and self-myofascial release), and pharmacology [9,10]. Of the above methods, in recent years much attention has been paid to compression therapy [11], in which the most frequent mention is External Pneumatic Compression (EPC) [12] as well as Intermittent Pneumatic Compression (IPC) [12]. This is especially seen in football, where over half of the players declared that the use of pressotherapy potentially accelerated regeneration [13].

Among the studies that used EPCs, a positive effect was found to increase flexibility and reduce muscle soreness (MS) [14,15], as well as reducing lymphoedema [16] and lactate [17]. The research conducted by Martin et al. (2015) showed that EPC did not statistically significantly affect the reduction of lactate after the 30-s Wingate test compared to the control group [17]. Similar relationships were found by Haun et al. (2017), in which they did not notice a statistical difference in muscle strength between the control group and the experimental group after resistance training in the form of back squats [11].

Using IPC has been reported to be effective in regeneration with short-term ECC efforts, reduction of fatigue [18], reduction of edema [19], improvement of local blood supply [20], and improvement in the ROM [12]. In subsequent studies, IPC was more effective at reducing high lactate levels than passive rest after exercise [21], and also statistically significantly reduced soft tissue stiffness after ECC training [19] and slightly reduced delayed post-exercise (Po-E) pain after short-term intense exercise [22].

Other studies have shown mitigating the effects of reducing muscle strength immediately after training [18] and improving the speed of a 400-m run [12].

This systematic review and meta-analysis aimed to examine the effectiveness of the pressotherapy to reduce DOMS after exercise. The primary endpoint was to assess pressotherapy the changes in MS and sports performance. The secondary endpoint is to establish the specific benefits on the selected outcomes of muscle functional capacities (e.g., strength, power), muscle damage markers (e.g., serum CK levels), joint ROM, and pain sensation.

## 2. Materials and Methods

The present review and meta-analysis were reported according to the Preferred Reporting Items for Systematic Reviews and Meta-Analyses (PRISMA) and follow the recommendations of the Cochrane Handbook for Systematic Reviews of Interventions [23]. The PRISMA 2020 statement: an updated guideline for reporting systematic reviews. Systematic Reviews [24].

### 2.1. Search Strategy and Screening Procedures

Searches were carried out on the following electronic databases: MEDLINE (PubMed and EBSCO), Web of Science, SPORT Discus. We did not have any limits and we searched all articles to March 2021 for studies aimed at determining the effect of pressotherapy on the magnitude and time course of Po-E muscle soreness and sports performance and recovery following exercise-induced muscle damage. We also searched current information about registers and reports in ClinicalTrials.gov. Additionally, we carried out a manual search of the bibliography of the included works and tracked their citations in the Scholar database. We head the same keywords as in databases. There were no associated publications, reports, or registers.

The search algorithm was conducted using PICO’s strategy [23] (type of studies, participants, interventions, comparators, and outcome assessment) and combined Medical Subject Headings, free-terms, and matching synonyms of the following related words: (1) population: healthy adults, “middle-aged”, “young adults”; (2) intervention: external assisted mechanical therapy, “external counterpulsation”, “lymphatic drainage”, “pressotherapy”, “intermittent pneumatic compression”, “pneumatic compression”, “pneumatic therapy”, “intermittent compression”, “compression therapy”, “compression massage”, “pneumatic massage”); (3) outcome: “Soreness”, “DOMS”, “inflammation”, “muscle fatigue”, “recovery”, “Delayed Onset Muscle Soreness”, “EIMD”, “hyperalgesia”, “allodynia”, “myalgia”; and (4) comparator: control conditions; RCT’s studies and cross-over. In addition, we searched the citations included in the identified publications deemed eligible for our study.

### 2.2. Inclusion Criteria

Those studies in which the title and abstract were related to the aim of the present review were included for full-text request. We included studies that (1) were conducted as randomized control trials (RCT) and cross-over designs; (2) included a mean participant age between 18 and 65 years old. (3) Healthy adults with exercise-induced muscle damage regardless of their level of sports activity and performance (4) were based on at least one exercise intervention described as “External assisted mechanical therapy” (machines).

### 2.3. Exclusion Criteria

Studies were excluded if (1) outcome measurements were not reported as DOMS max values, or (2) they were not written in English. A third reviewer (SW) resolved cases of initial reviewer disagreement. Nonrandomized experiments, observational studies, secondary studies (any types of evidence syntheses), and opinion pieces (e.g., narrative reviews, editorials) were excluded too.

### 2.4. Selection Process, Data Collection, Data Extraction, and Management

Two initial reviewers (MJ and MC) independently examined the titles and abstracts of retrieved articles to identify suitable studies and extracted the following information from the included studies: First author’s name and year of publication; study design; characteristics of the participants included; mean age; sample size and percentage of female subjects; weekly frequency, period and modality of External assisted mechanical therapy intervention; the reported measurement of Muscle functional capacities (e.g., strength, power), Muscle damage markers (e.g. serum CK levels), Joint ROM, and pain sensation. A third reviewer (SW) resolved cases of author disagreement.

### 2.5. Risk of Bias Assessment

The risk of bias of RCTs was assessed using the Cochrane risk-of-bias tool for randomized trials (RoB 2.0) [25], in which five domains were evaluated: Randomization process, deviations from intended interventions, missing outcome data, measurement of the outcome, and selection of the reported result. Each domain was assessed for risk of bias. Studies were graded as (1) “low risk of bias” when a low risk of bias was determined for all domains; (2) “some concerns” if at least one domain was assessed as raising some concerns but not at a high risk of bias for any single domain; or (3) “high risk of bias” when a high risk of bias was reached for at least one domain or the studied judgment included some concerns in multiple domains [24]. Assessment for individual randomized, parallel-group trials is presented in Figure 1. For pre-post studies and non-RCTs we used the Quality Assessment Tool for Quantitative Studies [25], in which seven domains were evaluated: Selection bias, study design, confounders, blinding, data collection methods, withdrawals, and dropouts. Each domain was considered strong, moderate, or weak. Studies were classified as “low risk of bias” if they presented no weak ratings; “moderate risk of bias” when there was at least one weak rating; or “high risk of bias” if there were two or more weak ratings [25]. Assessment for individually randomized, cross-over trials is presented in Figure 2. The risk of bias was independently assessed by two reviewers (MJ and PW). A third reviewer (SW) was consulted in case of disagreement.

### 2.6. Outcome Measures

Objective results of interest for meta-analyses from included baseline to last available follow-up. Data were typically collected immediately and 24 h, 48 h, 72 h, up to 96 h after the intervention.

### 2.7. Primary Outcomes

The primary endpoint was to assess the effect changes in MS and sports performance.

### 2.8. Secondary Outcomes 

The secondary endpoint was to assess muscle functional capacities (e.g., strength, power), muscle damage markers (e.g., serum CK levels), and joint ROM and pain sensation.

### 2.9. Statistical Considerations

Random-effects meta-analyses were performed using the Revman5.4.1 software [26]. Data was represented by Standardized Mean Difference and 95% Confidence Interval (CI). tau-squared Tau^2^, chi-squared Chi^2^ and I^2^ were used to investigate the presence of heterogeneity in meta-analysis. A *p* value < 0.05 was considered statistically significant. 

## 3. Results

### 3.1. Results of the Search

A total of 693 articles related to the topic were retrieved through a comprehensive database and other sources search, of which, 169 articles were duplicates. After removing all ineligible articles, a total of 12 RCTs were included in the analysis. The detailed screening process is shown in Figure 3.

### 3.2. Details of the Intervention Groups in the Included Studies

Characteristics of the included studies are summarized in Table 1.

There were five randomized controlled trials [18,29,30,32,33] and seven randomized crossover trials [11,12,19,27,28,31,34]. Overall, studies included patients from five countries: USA (*n* = 5), New Zealand (*n* = 3), Ireland (*n* = 1), Australia (*n* = 2), and Spain (*n* = 1).

The total study population of all selected articles comprised of 322 healthy volunteers with an unequal distribution of sex (n_male_ = 274; n_female_ = 48). Throughout all the studies, mean sample size ranged from 10 to 72 volunteers.

The average sample size of the pressotherapy group was 14.33 and the control group 13.25. The mean age of the study population was 28.1 years. In two studies the mean age was above 40 years. [1,2].

Two studies involved well-trained volunteers [11,33]. Three studies included runners [12,29,31]. One study included strength-trained males [30]. Two studies included physically active volunteers [18,27] and athletes [28,34], another two studies chose healthy participants [19,32]. Detailed information about the training status is presented in the Table 1.

### 3.3. Characteristics of the Exercise Protocols, Therapies and Outcomes

To induce muscle damage exercise protocols encompassed in running and other activities, five used run [11,12,28,29,31]. One of these types of exercise was sprint [28], another one was middle—6 km [11]—and three of the remaining five were long-distance run 62.7 [31]; 87.4 [31]; 102.8 [31]; 2 × 20 mile [29]; 161 km [12]. Two studies used back squats, 10 sets × 10 rep [30], and 10 sets of five repetitions [27]. Another way to induction DOMS intervention was ECC exercise on Biodex system [18], eccentric exercise performed with weight [19], plyometric exercise bout [32], countermovement jump (CMJ) [28], and wheelchair court sprints [33]. One study used specific training: Reverse grip battle rope waves, Farmers carry, Chin-ups, Bar hangs, Handgrip crushers [34]. Table 1 gives a detailed overview of the conducted exercise protocols.

Considerable variation was observed in therapy parameters among the studies. Intermittent sequential pneumatic compression (ISPC) was used in three studies [12,33,34]. Time of therapy was 2 min [2], 30 s/15 s [33], or 26 s/15 s [34]. External pneumatic compression (EPC) was used in three studies [11,27,28], two authors used the same parameters 70 mmHg inflation—30 s, deflation—30 s [11,27], and one study used 235 mmHg pressure [28]. The most popular therapy was IPC [18,19,29,31].

There was a different time of experimental and control condition; the majority performed therapy post-exercise, and after 24 h. The average therapy session was 30 min. The shortest time was 6 min [30] and the maximum was 1 h [11,27,29]. Total therapeutic exposition time varied from 20 to 30 min. [12,33,34] to longer times of 80 min to 6 h [18,19,31].

Outcome variables and time of measurement varied depending on the study. The period of measurement keeps on from Po-E [12,19,28,29,30,33,34] to 336 h after exercise [31]. The average time of access outcomes was 48 h. Muscle pain soreness and (CK) were the most-often measured. Six studies investigated CK [11,18,27,28,32], five MS [11,12,32,33,34], and eight pain Visual analogue scale (VAS) [11,12,19,28,29,30,31,32]. Other authors access Over Fatigue [12], Flexibility [11], Muscle Dynamometry and vertical jump (VJ) [18,19] C-reactive protein (CRP) [27,29], countermovement jump (CMJ), reactive strength index (RSI) [28,30,32], cortisol, testosterone, alpha-amylase, and immunoglobulin [28]. Detailed information about the measured parameters can be observed in Table 1.

Main effects were measured Po-E through to 336 h after. CK increased Po-E to 24 h [28], 72 h [18] and 168 h [27]. Haun (2017) concluded that after 168 h there was no significant change. Significant effect was observed after 24 h [18,28] and 96 h [11,27] and 120 h [27].

Muscle Pain increased Po-E to 24 h [28,30], 96 h [29], 120 h [19,31] and 168 h [12]. Significant effect was observed after one hour [30], 24 h [28,30,31], 48 h [29], 96 h [12]. In one study, an increase was observed Po-E to 144 h but with no significant changes [19].

Muscle soreness had a heterogeneous direction of changes. Some authors observed decreasing after exercise from 72 h to 144 h and significant changes were measured after 72 h and 120 h [11,31]. The majority observed significantly increasing MS Po-E and after 24 h to 96 h [12]. Velanzuela (2018) observed increasing MS after 24 and 48 h but without any significant changes [32]. Oliver (2021) observed increasing MS Po-E, post-recovery, and after 24 h and also without any significant changes [33]. Cranston (2020) observed increasing Po-E in all four muscle groups, post-recovery decreasing in three groups with significant differences between groups, and after 24 h increasing in all four muscle groups, with significant differences between groups [34].

Hoffman (2016) observed that muscle fatigue increases post-race, post-treatment significantly and reached significant difference between groups post-race 24–168 h [12]. Two other authors analyzed the change of these parameters [31,33] and Heapy (2018) observed changes post-race, 24–168 h, and 336 h after exercise, and post-race, the 24–72 h increase was significant [31]. Furthermore, there was a significant difference between the groups of 72 h, 96 h, and 120 h. In Oliver et al (2021) muscle fatigue Po-E, post-recovery, and 24 h Po-E remained unchanged [33].

Two studies assess muscle flexibility parameters [11,27]. Both observed increasing after 72 h and decreasing after 168 h. Swelling and stiffness were observed by Chleboun et al (1995) after 24–96 h and 120 h [19]. The stiffness increased after 24 and 48 h and then decreased to 120 h.

Two studies measured isometric strength [18,19]. Cochrane (2013) observed decreased peak isometric strength after 24 h and increased after 48 and 72 h—all changes were significant [18]. Chleboun (1995) observed a decrease after 24–72 h and an increase after 96 and 120 h [19].

Cochrane et al (2013) measured a few dynamometry parameters: Peak concentric 30°—decreased after 24, 48, and 72 h; peak concentric 180° decreased similar to previous parameters; peak ECC 30° and 180°—decreased after 24 h and increased after 48 and 72 h [18]. Other parameters: Average concentric 30°, 180° decreased after 24–72 h; average ECC 30°, 180° decreased after 24 h and increased after 48–72 h [18]. Northey et al. (2016) also measured concentric peak and he observed decreased post and after 1 h and then no significant changes [30].

Collins et al. (2019) assessed blood test results: cortisol, testosterone, immunoglobulin—increased Po-E and decreased after 24 h; Alpha-amylase—significant changes post and 24 h and between groups [28]. Oliver et al. (2021) measured blood lactate—post-recovery it decreased. C-reactive protein was measured in two studies [27,29] and remained unchanged after 24–144 h and 168 h [33].

Some authors used exercises to measure the main effect. Hoffman et al. (2016) and Heapy et al. (2018) used 400 m runs with increased time after 72 h [12] and 120 h [31], and decreased time after 120 h [12]. Another activity to measure effects was a 6 km run after 168 h Po-E. In a countermovement jump (CM) [28,30,32] heterogenous results were observed: decreased post and increased after 24 h—significant changes between groups [28]. Decreased post, 1 and 24 h post and 1 h showed significant changes [30]. After 24 h, it decreased and after 48 h there were no significant changes [32]. Valenzuela et al. (2018) also measured reactive strength index and had the same results as in the CMJ case [32]. Cochrane et al (2013) observed changes in vertical jump height—it decreased after 24 h and increased after 48 h and 72 h; vertical jump peak power—decreased after 24–72 h [18]. Northey et al. (2016) used squat jump (SJ) to measure the main effect and noted only decreased post and after 1 and 24 h [30]. Oliver et al. (2021) used a medicine ball throw test and wheelchair sprint on 5, 10, and 15 m, and observed decrease with post-recovery increase [33]. Sprint on every distance was increased. Cranston et al. (2020) used exercises: Grip strength dynamometer—decreased Po-E and post-recovery; Single-arm medicine ball throw—Po-E it decreased and then post-recovery increased; Max repetition single-arm preacher biceps curls—Po-E and recovery it decreased [34].

### 3.4. Subgroup Analysis

#### 3.4.1. Muscle Soreness

There was moderate and statistically significant reduction in MS in overall effect from 24 to 96 h after DOMS induction in pressotherapy intervention (Standard Mean Difference (SMD) = −0.33, 95% CI −0.49, −0.18; *p* < 0.0001; I^2^ = 7%). In the Subgroup 24 h Po-E (participants = 311; studies = nine) there was moderate but NS reduction in MS (SMD = −0.28, 95% CI −0.60, 0.04; *p* = 0.09; I^2^ = 43%), 48 h Po-E (participants = 144; studies = nine) there was moderate and significant reduction in MS (SMD = −0.40, 95% CI −0.73, 0.07; *p* = 0.02; I^2^ = 0%), 72 h Po-E (participants = 124; studies = four) there was moderate but NS reduction in MS (SMD = −0.37, 95% CI −0.79, 0.05; *p* = 0.08; I^2^ = 24%) and 96 h Po-E (participants = 124; studies = four) there was moderate but NS reduction in MS. In overall effect from 24 to 96 h heterogeneity was small (I^2^ = 7%; χ^2^ = 22.6, df = 21; *p* = 0.96). Only in the subgroup 24 h Po-E we detected NS heterogeneity (I^2^ = 43%; χ^2^ = 14.16, df = 8; *p* = 0.08). After48–96 h, heterogeneity was low. Subgroup analysis from 24 h to 96 h did not reveal a statistically significant difference (*p* = 0.96) (Figure 4).

#### 3.4.2. Jump Performance

In 24 h Po-E (participants = 84; studies = 4; SMD = −0.05, 95% CI −0.47, −0.38; *p* = 0.99; I^2^ = 0%), 48 h Po-E (participants = 40; studies = 2; SMD = −0.01, 95% CI −0.61, 0.63; *p* = 0.77; I^2^ = 0%), and 72 h Po-E (participants = 20; studies = 1; SMD = −0.10, 95% CI −0.98, 0.78; *p* = 0.82; I^2^ = not applicable) there was a small statistically NS effect of pressotherapy on jump height. In overall effect from 24 to 72 h (SMD = −0.04, 95% CI −0.36, −0.29; *p* = 0.82) heterogeneity was small (I^2^ = 0%; χ^2^ = 0.25, df = 21; *p* = 1.00).

Subgroup analysis from 24 h to 96 h did not reveal a statistically significant difference (*p* = 0.98) (Figure 5).

#### 3.4.3. Creatine Kinase

There was an NS increase in serum CK activity in overall effect from 24 to 96 h after DOMS induction in pressotherapy intervention (SMD = 0.41, 95% CI −0.07, 0.89; *p* = 0.09; I^2^ = 63%). In the subgroup 24 h Po-E (participants = 81; studies = four; SMD = 0.14, 95% CI −0.30, 0.58; *p* = 0.54; I^2^ = 0%), 48 h Po-E (participants = 60; studies = three; SMD = 0.52, 95% CI −0.77, 1.81; *p* = 0.43; I^2^ = 82%), 72 h Po-E (participants = 40; studies = two; SMD = 0.49, 95% CI −1.25, 2.23; *p* = 0.58; I^2^ = 85%) there were small (24 h) and moderate (48–72 h) but NS increases in serum CK activity. In the 96 h Po-E group (participants = 20; studies =one) there was large and significant increase in CK activity for the pressotherapy group (SMD = 1.26, 95% CI 0.28, 2.23; *p* = 0.01; I^2^ = not applicable).

Overall, the heterogeneity in effects from 24 to96 h was moderate (I^2^ = 63%; χ^2^ = 24.47, df = 9; *p* = 0.004). Only in the subgroup 24 h Po-E we detected homogeneity (I^2^ = 0%; χ^2^ = 2.44, df = 3; *p* = 0.49). 48 h (I^2^ = 82%; χ^2^ = 11.05, df = 2; *p* = 0.004) and 72 h (I^2^ = 85%; χ^2^ = 6.78, df = 1; *p* = 0.009) heterogeneity was large. Subgroup analysis from 24 h to 96 h did not reveal a statistically significant difference (*p* = 0.23) (Figure 6).

## 4. Discussion

### 4.1. Brief Study Informations: Purposes, Direction, and Possible Main Outcomes

Most of the studies used a one-time protocol to assess the time of post-workout regeneration. The most reliable method would be to use it multiple times under different conditions to maximize result accuracy [35].

The best methods of post-workout recovery are sleep and a proper diet [36,37]. Additional methods can only be supplementary. For the assessment of the credibility of the studies, we recommend that the information on whether pressotherapy was the primary method or an addition to the more comprehensive scheme should be included in the research Methodology section.

Maximizing the efficiency of post-workout adaptation is crucial for athletes to maintain an appropriate performance level throughout the season and during the pre-competition preparation periods [38,39]. This is especially important in sports with a high frequency of competitions (i.e., team sports such as soccer and basketball), as well as in disciplines where the athlete prepares for a long time for one event in which their organism achieves peak performance (i.e., individual disciplines such as sprinting or swimming).

We stipulate that pressotherapy does NS affect post-workout regeneration and can only supplement a complex protocol.

### 4.2. Serum CK Level

The blood level of CK is an indicator of the status of muscle damage and of change in both pathological and normal conditions [40]. An increase in this enzyme may predict a state of microscopic tissue impairment after acute and prolonged injuries. Variables in CK level are also observed under physiological conditions in athletes after demanding training. The highest CK growth is observed after prolonged exercise, i.e., triathlon events and demanding strength exercises, or activities that include an eccentric muscle contraction phase, i.e., downhill running [41,42]. In our study, we saw an improvement in this parameter, which suggests that pressotherapy improves regeneration. However, its impact was not statistically significant in any case except the 96 h Po-E group, which had the lowest number of participants. In addition, a significant result was observed in the longest period after the training was performed, which leaves some ambiguity as CK activity decreases with time and it is a natural process [43]. Not without relevance is also the fact that a significant result was observed by Haun et al., who investigated CK levels on a group of trained high-volume endurance athletes, who underwent over 70 h of exertion per week for 3 months. Although significant results have been observed, previous studies suggest that CK levels naturally decline between days 4 and 10 after exercise [44]. The characteristics of the test group (endurance athletes) and testing protocol could also affect the results, as resting CK levels are higher in the trained population [45,46] and everyday strenuous workouts may cause persistent blood rise of CK [47]. Therefore, the potential outcome of pressotherapy on a different group of people would not be so important. To summarize, in the current state of knowledge, pressotherapy should not be recommended as the basic method of recovery after exercise, because there is a large heterogeneity of previous research results.

### 4.3. DOMS

DOMS is a regular experience for advanced or beginner athletes. Its manifestations can range from muscle stiffness to severe excruciating pain [48]. DOMS is most prevalent at the beginning of the sporting season when athletes are returning to training following a period of reduced activity [49]. DOMS is also common when athletes are first introduced to certain types of activities regardless of the time of year. DOMS can negatively attenuate athletic performance [50]. Possible mechanisms include a reduction in joint ROM, peak torque, and a feeling of pain [48]. Compensation methods may raise the probability of further injury [51,52] when participants try to return to activity too early without completing the full recovery process. Therefore, it is of high importance to search for new methods of the most effective regeneration and reduction of MS. Commonly described in the literature are pressotherapy [48], stretching [53], cryotherapy [54], and massage, mainly considered as self-foam rolling. It has been the most often assessed parameter in selected studies. Although pressotherapy is one of the methods of DOMS reduction, our results indicate that its use for this purpose remains questionable. Only when MS was measured after 48 h, was a a significant effect of pressotherapy observed. This method also significantly alleviates DOMS when considering the whole population and all protocols. On the other hand, no significant reduction in MS was found in the remaining groups. Taking into account the previously mentioned methods of therapy, which are easily available (stretching or foam-rolling), as well as low-cost (cryotherapy and water immersion) or self-applicable and physiologic (i.e., rest), there are few arguments in favor of the wide use of pressotherapy in the current state of knowledge. High prices and limited availability suggest other forms as a method of choice and first-line treatment strategy. However, pressotherapy has shown some positive effects, mainly limited to the 48 h Po-E period, so while the above-mentioned factors are not a barrier, it can be used in some circumstances [55] (e.g., in professional athletes as a supplemental method).

### 4.4. Jump Performance

The level of muscle power in the lower limbs is a vital factor in numerous disciplines, such as sprinting [56,57] or in decisive moments of team sports [58,59]. In a widespread view, the research has demonstrated that jump heigh is an applicable index to characterize power output, mainly described by the association found between them [60]. It is meaningful that upright jump may be easily evaluated and hereafter used by team staff and physical trainers to categorize the level of athletes’ muscle power within a wider group of participants [61,62]. Due to the great practical importance of jump performance in the overall assessment of an athlete’s fitness and the development of motor skills, it is crucial to properly place this type of activity in the training plan and the microcycle [63,64]. Effective recovery after jumping efforts would be of key importance, hence the influence of pressotherapy on jump performance was also assessed in this meta-analysis. In our review, we did not observe any significant effect of pressotherapy on jump ability performed at various intervals from the previous exercise. Further investigation is needed to specify whether and in what population this method will be an effective approach for improving jump performance and overall power generation.

### 4.5. Practical Implications

This study has several practical implications and contributes significantly to the actual state of knowledge in this research area. It can be used by motor preparation specialists and physiotherapy professionals in the prescription of individualized, advanced recovery strategies. This is especially important when maximizing the effectiveness of post-exercise regeneration is necessary (e.g., for elite athletes during the beginning of the season or directly before competitive event).

## 5. Conclusions

The conducted systematic review and meta-analysis assessed 12 randomized controlled studies investigating the outcome of pressotherapy on the recovery of absolute (i.e., physiological), and subjective (i.e., perceptual) outcomes. The findings indicate only moderate benefits of using pressotherapy as a recovery intervention, dependent on the type of exercise and used protocol. A reduction in DOMS, changes in CK level, and improvements in perceived recovery were observed after pressotherapy, although they were usually not significant. Dose–response relationships emerged for several variables indicating that different duration protocols may improve the efficacy of pressotherapy if applied after exercise. We recommend further, and continuing, research on various populations and broadening tested protocols to obtain the highest possible homogeneity of results and to facilitate the creation of a consensus statement on whether pressotherapy seems to be an effective method in minimizing exercise-induced negative effects.

## 6. Limitations

Although, this paper has a few limitations. Firstly, we performed a comprehensive literature investigation, where we excluded articles that were not published in English. However, from an actual point of view, we suppose this will have a minor effect on our outcomes [65]. Nevertheless, we conducted a reasonable attitude to overwhelm these barriers and attempted to stick to principles of open science. Secondly, the protocols used and the study groups differed between the selected articles. Third, the time of outcome evaluation from the preliminary endpoint was not identical in all trials. Fourth, the particular subgroup analyses were conceivably underpowered due to their small participant number and should be interpreted carefully. To enhance the validity of results in similar research, future randomized studies should concentrate on better conducting and reporting of applied protocol and methodology, intention-to-treat examination, assessor blinding, random sequence generation, control group observation, and reporting of adverse events or the possible other influencing factors. Moreover, not all databases (i.e., EMBASE) were searched.

## Figures and Tables

**Figure 1 jcm-11-02077-f001:**
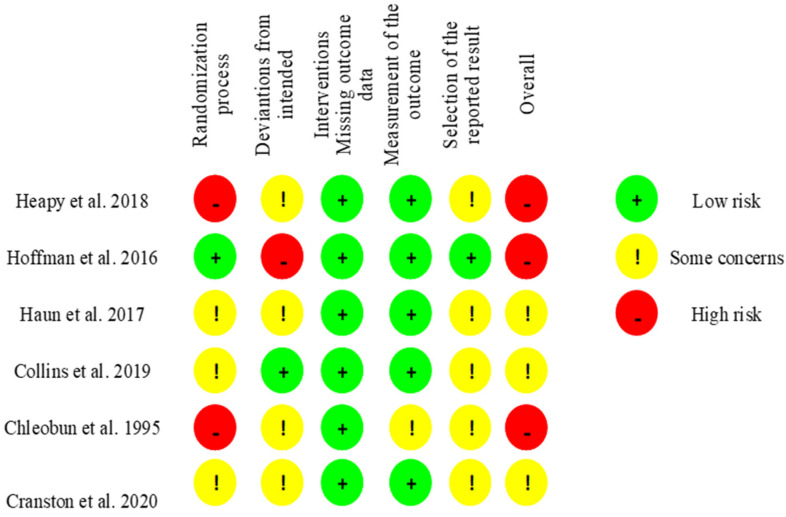
Risk of bias 2 tool. Assessment for individual randomized, parallel-group trials [25].

**Figure 2 jcm-11-02077-f002:**
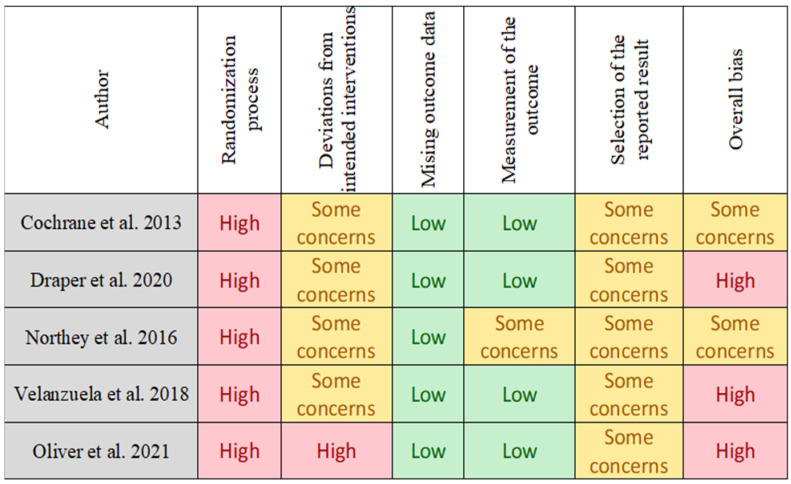
Risk of bias 2 tool. Assessment for individually randomized, cross-over trials [25].

**Figure 3 jcm-11-02077-f003:**
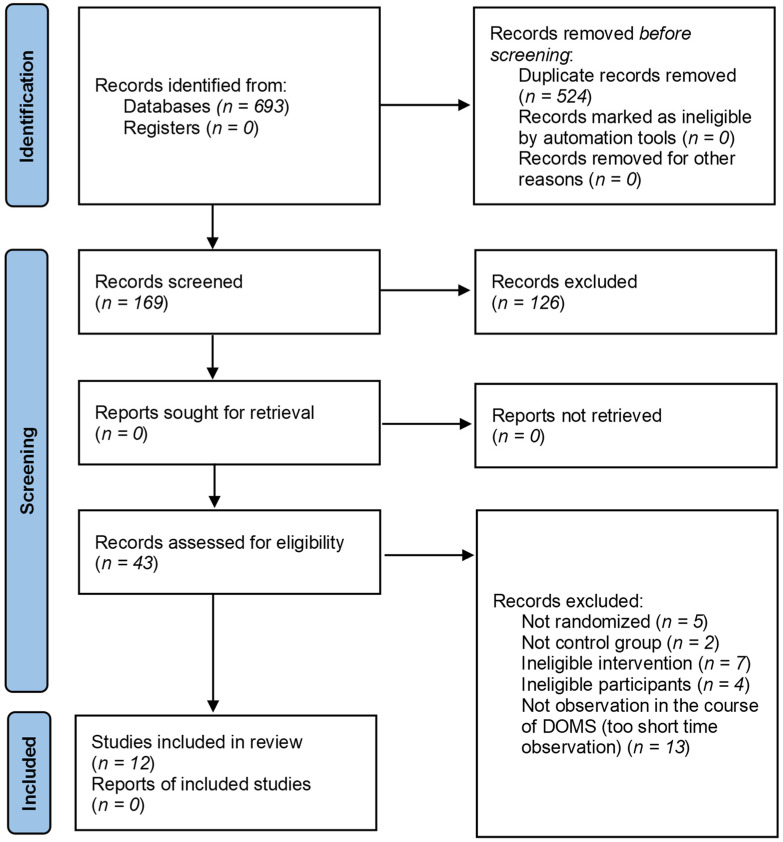
PRISMA flow diagram of included/excluded studies.

**Figure 4 jcm-11-02077-f004:**
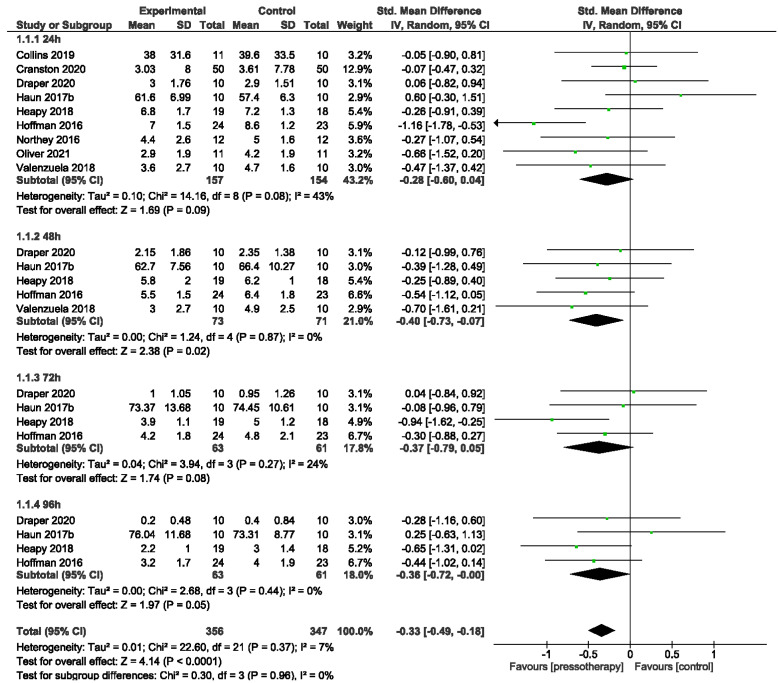
Effects of pressotherapy on muscle soreness from 24 h to 96 h after exercise.

**Figure 5 jcm-11-02077-f005:**
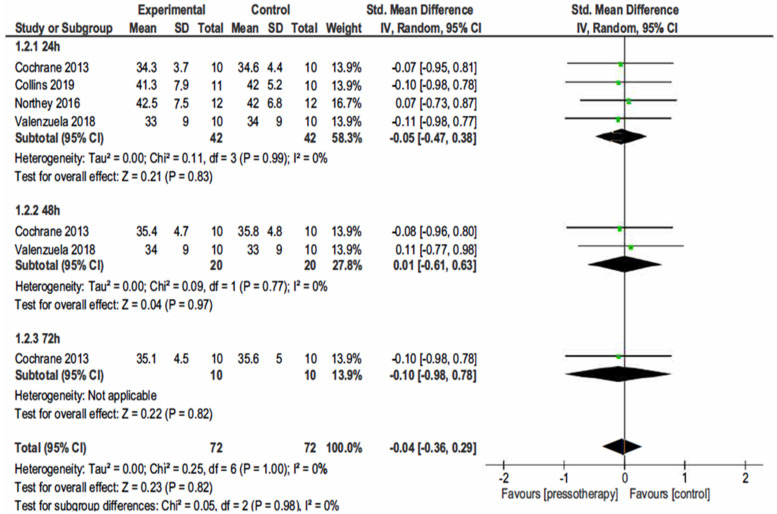
Effects of pressotherapy on jump performance from 24 h to 96 h after exercise. SMDs are calculated from CMJ, VJ, etc.

**Figure 6 jcm-11-02077-f006:**
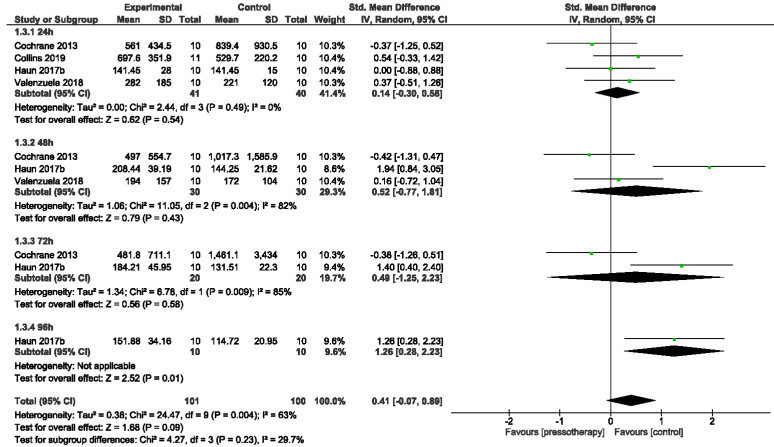
Effects of pressotherapy on serum CK activity from 24 h to 96 h after exercise.

**Table 1 jcm-11-02077-t001:** The key characteristic of selected studies (*n* = 12).

Author/Country	Design/Publication Year	Participant Cohort (Training Status, Sex, Age)	Sample Size (*n*)	Experimental vs. Control Condition	DOMS Induction Intervention	Outcome Variables and Time of Measurement Post-Exercise (hrs)	Main Effects [* *p* < 0.05: Pre-Post (× Time)]	Total Exposition Time	Therapy Parameters
Hoffman et al. [12]/USA	RCT/2016	participants in the 2015 161-km Western States Endurance Run, men (IPC:43 ± 8 years, Massage:46 ± 10 years, con.:45 ± 9 years)	*n* = 72 *n* = 24 exp. (IPC) *n* = 25 exp. (Massage) *n* = 23 con.	45 min post exercises IPC (20 min), 45 min post-exercise Massage (20 min) vs. Placebo therapy (20 min)	161 km ultramarathon race	400-m run times, Muscle Pain and Soreness, Overall Fatigue (prerace, postrace, posttreatment, 24–168 h post-race day)	400-m run times (pre↔, post 72 h↑, 120 h↓) Lower-Body Muscle Pain and Soreness (pre↔, postrace↑*, posttreatment↑*#, post 24–96 h↑*, post-120–168 h↑ Time and interaction effect* (no group effect) Muscular Fatigue (pre↔, postrace↑, post-treatment↑*#, postrace 24–168 h↑) Time and interaction effect* (no group effect)	20 min ISPC 20 min Massage 20 min Con.	ISPC—80 mmHg Massage—(the 30 s—calf and hamstring, 1 min—quadriceps), compression (2 min—calf and quadriceps, 3 min hamstring), tapotement (30 s leg and quadriceps)
Haun et al. [27]/USA	RCT/2017	endurance-trained male, participating in ≥72 h per week of endurance exercise for at least 3 months. (EPC:21 ± 0.4 years, con:21.1 ± 0.6 years)	*n* = 18 *n* = 9 exp. (EPC) *n* = 9 con.	24 h, 48 h, 72 h post-exercises EPC (1 h) vs. Placebo therapy (1 h) 96 h, 120 h treatments only EPC (1 h) vs. placebo therapy (1 h)	6 km run on the treadmill at an incline of 1% (pre and 16 h)	CK, Muscle Pain, and Soreness (pre-exercises, 72 h to 168 h), Flexibility (pre-exercises, 72 h to 168 h), 6-km run times (pre-exercises, 168 h)	CK (pre, 72 h↑, 96 h↑*, 120 h↑, 144 h↑, 168 h↔) Time effect* (No group or group × interaction effect) Muscle Soreness (pre, 72 h↓*, 96 h↓, 120 h↓*, 144 h↓, 168 h↔) Time effect* (No group or —group effect) Flexibility (pre, 72 h↑, 96 h↔, 120 h↔, 144 h↔,168 h↓) 6 km run time (pre, 168 h↓)	300 min EPC 300 min EPC Con	EPC—70 mmHg (inflation—30 s/deflation—30 s)
Cochrane et al. [18]/NZ	RCO/2013	10 healthy males, involved in physical activity (21.0 ± 1.7 years)	*n* = 10 *n* = 10 exp. (IPC) *n* = 10 con.	Immediately post-exercises, 24 h post-exercise, 48 h post-IPC (30 min) vs. Placebo therapy (30 min)	3 sets × 100 rep. strenuous bout of eccentric exercise on BIODEX	CK, VJ, Muscle Dynamometry ISO 75° - CON 30°/s; 180°/s - ECC 30°/s; 180°/s) (Pre, 24 h, 48 h, post 72 h)	CK (pre, 24 h↑*, 48 h↑, 72 h↑) VJ height (pre, 24 h↓, 48 h↑, 72 h↑) VJ peak power (pre, 24 h↓, 48 h↓, 72 h↓) Peak ISO (pre, 24 h↓*, 48 h↑*, 72 h↑*) Peak CON 30° (pre, 24 h↓*, 48 h↓, 72 h↓) Peak CON 180° (pre, 24 h↓, 48 h↓, 72 h↓) Peak ECC 30° (pre, 24 h↓, 48 h↑, 72 h↑) Peak ECC 180° (pre, 24 h↓, 48 h↑, 72 h↑) Ave ISO 75° (pre, 24 h↓, 48 h↑, 72 h↑) Ave CON 30° (pre, 24 h↓*, 48 h↓, 72 h↓) Ave CON 180° (pre, 24 h↓, 48 h↓, 72 h↓) Ave ECC 30° (pre, 24 h↓, 48 h↑, 72 h↑) Ave ECC 180° (pre, 24 h↓, 48 h↑, 72 h↑)	90 min IPC 90 min Con	IPC—cell 1 (distal)—70 mmHg, cells 2–4 80 mmHg, cell 5 (proximal) 60 mmHg/deflation—30 s.
Collins et al. [28]/IE	RCT/2019	21 male team sport athletes (21.6 ± 3.4 years)	*n* = 21 *n* = 11 exp. *n* = 10 con.	Pre-, post-, 24 h post- exercises ECP (20 min) vs. Placebo therapy (20 min)	Max CMJ, 2 × 20 sprint, and second max CMJ	CK, C, T, IgA, sAA, VAS, CMJ height (Pre, post, 24 h post)	CK (pre, post↑*, 24 h↑*) Main effect for time* Cortisol (pre, post↑, 24 h↓) Testosterone (pre, post↑*, 24 h↓*) Main effect for time Alpha-Amylase (pre, post↑*#, 24 h↑*#) Main effect for time, and group Immunoglobulin—A (pre, post↑, 24 h↓) VAS (pre, post↑, 24 h↑*) Main effect for time CMJ (pre, post↓*#, 24 h↑*#)	60 min ECP 60 min Con	ECP—235.3 ± 26.9 mmHg
Draper et al. [29]/USA	RCO/2020	10 runners, endurance-trained males (38.7 ± 11.2 years)	*n* = 10 *n* = 10 exp. *n* = 10 con.	1 h, 24 h, 48 h, 72 h, 96 h, 120 h post- IPC (1 h) vs. 1 h, 24 h, 48 h, 72 h, 96 h, 120 h post- Placebo therapy (1 h)	2 × 20 mile runs at 70% VO2 max separated by 3 or 4 weeks	CRP, VAS (pre, post, and 24 h, 48 h, 72 h, 96 h, 120 h post)	CRP (pre-, post-run ↔, 24 h↑*, 48 h↑, 72 h↑,96 h↔120) Main effect of time VAS (pre, post-run↑*, 24 h↑*, 48 h↑*, 72 h↑,96 h↑, 120 h↔ pre-run)	6 h IPC 6 h Con	IPC—90 mmHg for cell 1 (distal) and cell 5 (proximal) and 100 mmHg for cells 2–4 (compression 30 s)
Northey et al. [30]/AU	RCO/2016	12 strength-trained male (24.0 ± 6.3 years)	*n* = 12 *n* = 12 exp. *n* = 12 con.	1 h post-exercises SIPC (45 min) vs. Placebo therapy (45 min)	10 sets × 10 rep. of back squats at 70% 1 repetition maximum	VAS, CON (peak of quadriceps), SJ, CMJ (Pre, post, 1 h, 24 h)	CON peak (pre, post↓*, 1 h↓*, 24 h↔) SJ (pre, post↓*, 1 h↓*, 24 h↓*) CMJ (pre, post↓*, 1 h↓*, 24 h↓) VAS (pre, post↑*, 1 h↑*, 24 h↑*)	12 min OCC -2 sets × 3 min (per leg) 45 min SIPC 45 min Con	SIPC—80 mmHg (deflation—15 s) OCC 220 mmHg (inflation 3 min)
Heapy et al. [31]/NZ	RCT/2018	56 ultramarathoners (con. = 19; 42 ± 9 years), (IPC = 18; 41 ± 8 years), (Massage = 19; 43 ± 9 years), men	*n* = 56 *n* = 18 exp. (IPC) *n* = 19 exp. (Massage) *n* = 19 con.	Post-race, 24 h, 48 h, 72 h post-race IPC (20 min) post-race, 24 h, 48 h, 72 h post-race Massage (25 min) vs. Placebo therapy (20 min)	Run race—three distance options of 62.7 km, 87.4 km, and 102.8 km	400 m run times (pre-race 1, pre-race 2, post-race at 72 h, 120 h, 168 h, and 336 h), VAS, Fatigue Scores (pre, post, day 24–168 h post and 336 h post)	400 m run times (pre-race 1, pre-race 2↔, 72 h↑, 120 h↑, 168 h↔, 336 h↔) Time effect* (No group, or interaction effect) VAS (pre-race, post-race↑*, 24 h↑*, 48 h↑, 72 h↑,96 h↑, 120 h↑, 144 h↔, 168 h↔, 336 h↔) Time effect* (No group or interaction effect) Muscle Fatigue (pre-race, post-race↑*, 24 h↑*, 48 h↑*, 72 h↑*#, 96 h↑#, 120 h↑#, 144 h↔, 168 h↔, 336 h↔) Time and interaction effect* (No group effect)	80 min IPC 100 min Massage 80 min Con.	IPC—80 mmHg
Chleboun et al. [19]/USA	RCT/1995	22 college women students (21.7 ± 0.7 years)	*n* = 22 *n* = 22 exp. (IPC) *n* = 10 con. (passive rest)	Post-exercise, 24 h, 48 h, 72 h, 96 h, 120 h post IPC (20 min) vs. Placebo therapy (20 min)	3 sets of ECC exercise performed with weights equal to 90%, 80%, and 70% of the ISO MVC	Pain (five-point pain-rating scale), Swelling (post, day 1 to 5), Stiffness, and Isometric Strength (pre-exercise, pre-, post-IPC days 1 to 5)	Pain (post, 24 h↑, 48 h↑, 72 h↑, 96 h↑, 120 h↑) Swelling (post, pre IPC (post IPC), 24 h↑ (24 h↑*), 48 h↑ (48 h↑*), 72 h↑ (72 h↑*), 96 h↑ (96 h↑), 120 h↑ (120 h↑*)) Stiffness (post-, pre-IPC, (post-IPC), 24 h↑ (24 h↑), 48 h↑ (48 h↓*), 72 h↓ (72↓*), 96 h↓ (96 h↓), 120 h↓ (120 h↓)) Strength (post-, pre-IPC (post-IPC) 24 h↓ (24 h↓), 48 h↓ (48 h↓), 72 h↓ (72 h↓), 96 h↑ (96 h↑), 120 h↑ (120 h↑))	120 min IPC	IPC—60 mmHg (inflation 40 s/deflation 20 s)
Velanzuela et al. [32]/ES	RCO/2018	10 healthy participants (27 ± 4 years), 7 men, 3 females	*n* = 10 *n* = 10 exp. *n* = 10 con.	Post-exercises, 24 h post-EECP (30 min) vs. Placebo therapy (30 min)	Plyometric exercise bout (10 sets of 10 jumps)	Muscle Soreness (VAS), CK, CMJ, RSI (pre and 24 and 48 h post)	Muscle Soreness (pre, 24 h post↑, 48 h post↑) CK (pre, 24 h post↑, 48 h post↑) CMJ (pre, 24 h post↓, 48 h post↔) RSI (pre, 24 h post↓, 48 h post↔)	60 min EECP 60 min Con.	EECP—80 mmHg
Haun C.T. et al. [11]/USA	RCT/2017	20 resistance-trained male (21.6 ± 2.4 years)	*n* = 10 *n* = 10 exp. (EPC) *n* = 10 con.	48 h, 72 h, 96 h, 120 h, 144 h post-EPC (1 h) vs. Placebo therapy (1 h)	10 sets of five rep. at 80% of back squat 1 RM	CK, Flexibility (pre, 48–168 h post) CRP (pre, 8–168 h post)	CK (pre, 72 h↑*, 96 h↑*, 120 h↑*, 144 h↑, 168 h↑) Flexibility (pre, 72 h↑*#, 96 h↑, 120 h↑*, 144 h↑, 168 h↓) CRP (pre, 48 h↑, 72 h↑, 96 h↑, 120 h↑, 144 h↑, 168 h↑)	5 h EPC 5 h Con.	EPC—70 mmHg (inflation—30 s/deflation—30 s)
Oliver et al. [33]/NZ	RCO/2021	11 well-trained wheelchair basketball and rugby athletes (33 ± 10 years), men	*n* = 11 *n* = 11 exp. *n* = 11 con.	post exercises ISPC (20 min) vs. Placebo therapy (30 min)	10 wheelchair court sprints (28 m). Ten times figure of eight agility drill (the 30 s). Ten sprints (28 m) immediately followed by three medicine ball chest throws	Medicine Ball Throw (m), Wheelchair Sprint, 5, 10, 15 (m) (pre-ex, post-ex, post-rec) Muscle Soreness 0–10 scale and Muscle Fatigue 0–10 scale (pre-ex, post-ex, post-rec, 24 h post-rec) Blood Lactate (post-ex, post-rec)	Medicine Ball Throw (pre-ex, post-ex↓, post-rec↑), Wheelchair Sprint: (5 m) (pre-ex, post-ex↑, post-rec↑) (10 M) (pre-ex, post-ex↑, post-rec↑) (15 m) (pre-ex, post-ex↑, pot-rec↑) Muscle Soreness (pre-ex, post-ex↑, post-rec↑, 24 h post↑) Muscle Fatigue (pre-ex, post-ex↑, post-rec↑, 24 h post↑) Blood Lactate (post-ex, post-rec↓)	20 min ISPC 30 min Con.	ISPC—80 mmHg (inflation 30 s/deflation 15 s)
Cranston et al. [34]	RCT/2020	50 resistance-trained athletes (27 ± 4 years), 37 men, 13 females	*n* = 50 *n* = 25 exp. *n* = 25 con.	post exercises ISPC (30 min) vs. Placebo therapy (30 min)	Fatiguing Exercise Circuit (consisted of five different exercises): 1. Reverse grip battle rope waves (the 60 s) 2. 20 m Farmers carry (20 kg for women and 30 kg for men) 3. Chin-ups (maximum number of repetitions) 4. Chin-up bar hangs (long as possible with their hands in a pronated grip) 5. Handgrip crushers (as many times as possible)	Grip Strength Dynamometer (kg), Single-Arm Medicine Ball Throw (m), Preacher Bench Bicep Curls- max repetitions (pre-ex, post-ex, post-rec)	Grip Strength Dynamometer (pre-ex, post-ex↓, post-rec↓) Single-Arm Medicine Ball Throw (pre-ex, post-ex↓, post-rec↑) Max. Rep. Single-Arm Preacher Bench Bicep Curls (pre-ex, post-ex↓, post-rec↓) Triceps Brachii Long Head Soreness (pre-ex, post-ex↑, post-rec↑#, 24 h post-rec↑#) Biceps Brachii Soreness (pre-ex, post-ex↑, post-rec↓#, 24 h post-rec↑#) Extensor Digitorum Soreness (pre-ex, post-ex↑, post-rec↓#, 24 h post-rec↑#) Flexor Carpi Radialis Soreness (pre-ex, post-ex↑, post-rec↓#, 24 h post-rec↑#)	30 min ISPC 30 min Con.	ISPC—80 mmHg (inflation—26 s/deflation—15 s)

Abbreviations: PCD (pneumatic compression device), CS (compression sleeve), PC (pneumatic compression), EPC (external pneumatic compression), ECP (External counterpulsation), EECP (Enhanced external counterpulsation), IPC (intermittent pneumatic compression), ISPC (intermittent sequential pneumatic compression), OCC (evaluate vascular occlusion), SIPC (sequential intermittent pneumatic compression), VJ (vertical jump), SJ (squat jump), CK (creatine kinase), LDH (lactate dehydrogenase), ISO (isometric), CON (concentric), ECC (eccentric), HIIT (high intensity interval training), HIE (high-intensity exercise), CMJ (countermovement jump), DEC (deceleration), AMRAP (as much repetitions as possible), ALAP (as long as possible), WAnT (Wingate anaerobic test), THB (total hemoglobin), O2HB (oxyhemoglobin), HHB (deoxyhemoglobin), ROM (range of motion), C (cortisol), T (testosterone), IgA (immunoglobulin-A), sAA (salivary alpha-amylase), CRP (C-reactive protein), PkP (peak power), AP (average power), FI (fatigue index), BLa (blood lactate concentration), NRS (numeric rating scale), CWI (cold water immersion), MuscleMechFx (muscle mechanical function), RPE (rate of perceived exertion), DM (Muscle radial deformation), TC (time of contraction), BF (biceps femoris), RF (rectus femoris), RSI (reactive strength index). #—significant difference between groups, * *p* < 0.05, ↑—significant increase, ↓—significant decrease, ↔—no significant change.

## Data Availability

Data are available from the corresponding author upon reasonable request.

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
