# Peer review of "The Effect of Pressotherapy on Performance and Recovery in the Management of Delayed Onset Muscle Soreness: A Systematic Review and Meta-Analysis"

_jcm, 2022, doi:10.3390/jcm11082077_

Round 1

Reviewer 1 Report

Thank you for the opportunity to review the manuscript. Some points need to be further clarified. Are they:

1. Make the objectives of the manuscript clearer.
2. Was the review protocol previously recorded?
3. How were these research bases chosen? For example, why was EMBASE not consulted?
4. Regarding PICO's strategy. What is the reason for choosing normal individuals? The resource used is after effort and mostly in athletes.
5. There is clear heterogeneity among study participants. Even the types of studies included. (RCT) and cross–over designs.

In general, the systematic review. It has a wide feature. Heterogeneity of studies and subjects included in the study. Much for published reviews on the topic and for the heterogeneity of the studies and subjects included in this review. Also, due to the reduced spectrum of bases used.

Author Response

Dear reviewer,

Thank you very much for the thorough analysis of our manuscript, for your valuable and helpful comments and for giving us the opportunity to revise and improve our submission. We hope that our replies and explanations, as well as the amendments to the manuscript, fully address your concerns.

In the following, please find our answers to your comments. 

We conducted a systematic review in accordance with PRISMA guidelines. It is the first systematic review in this field in which a numerical synthesis in the form of a meta-analysis was included.

.

  1. Make the objectives of the manuscript clearer

Response: Thank you for the suggestion, we tried to improve the objectives.

  1. Was the review protocol previously recorded?

Response:

Review was registered online in PROSPERO (Date of registration in PROSPERO; 13 December 2020), CRD42020189382 Available from: https://www.crd.york.ac.uk/prospero/display_record.php?ID=CRD42020189382

  1. How were these research bases chosen? For example, why was EMBASE not consulted?

Response: The use of additional manual search for potential papers was carried out by analyzing the bibliography of the papers included in our review as well as by following the citations of the included papers in the Scholar database. The use of Scholar, MEDLINE (PubMed and EBSCO), Web of Science, SPORT Discus databases gives enough coverage for searches. In accordance with the question, we additionally conducted a search (20 March 2022) in the EMBASE database and after excluding duplicates, no additional literature items meeting the inclusion criteria were identified. We updated this information in the manuscript.

  1. Regarding PICO's strategy. What is the reason for choosing normal individuals? The resource used is after effort and mostly in athletes.

Response: The PRISMA framework was the method used to conduct the systematic review and meta-analysis, and the search strategy was based on the population, intervention, control and outcome (PICO) model. Assumptions are made in the inclusion criteria. We want to investigate the effect of using the intervention (pressotherapy) in people suffering from post-exercise muscle soreness (population), which is an unpleasant experience that increases fatigue and reduces functional abilities. The comparison of changes was carried out in relation to people not receiving the intervention (control).

  1. There is clear heterogeneity among study participants. Even the types of studies included. (RCT) and cross–over designs.

Response:

In the manuscript, we included both cross-over and parallel studies in the analysis. Such a possibility is permissible in situations where the carry-over effect in cross-over trials is low. Sufficient washout time in all cross over trials was used. Due to the nature of the intervention, the use of a placebo is not possible and effective blindness of the subjects or researchers is very limited, which was determined in the assessment of bias. In the statistical evaluation, the heterogeneity of the data was at a low level and the random effects meta-analysis model used in the calculations enables the correct assessment of the effect size.

Once more, we would like to thank the Reviewer for the important comments.

Yours sinserely,

Szczepan Wiecha on behalf of all authors.

Reviewer 2 Report

The authors present a good research paper. 

  • The relevance of the topic: Good.
  • Introduction: Good.
  • Methodology: Can be improved.
  • Results: Good.
  • Discussion: Good.     

In general, the paper follows an adequate structure and correct scientific support and can be published considering some limitations. The work is interesting in the field of Training performance, recovery, and physiological properties. However, there are a series of limitations that should be considered.

In the first place, carry out a review of the existing literature related to the subject, being essential to inquire into the MPDI – International Journal of Envionmental Research and Public Health journal itself, since there are papers related to its manuscript that can help to improve it. Therefore, include those references, if any, especially from the last five years. In addition, recommend reading some papers related to the topic of Football:

Palmieri, B., Palmieri, L., Mambrini, A., Pepe, V., & Vadalà, M. (2021). Onco-Esthetics Dilemma: Is There a Role for Electrocosmetic-Medical Devices?. Frontiers in Oncology, 2149.

Salinas-Huertas, S., Luzardo-González, A., Vázquez-Gallego, S., Pernas, S., Falo, C., Pla, M. J., ... & García-Tejedor, A. (2021). Risk factors for lymphedema after breast surgery: A prospective cohort study in the era of sentinel lymph node biopsy. Breast Disease, (Preprint), 1-12.

Pervykh, S., & Bychkova, N. (2022). Effect of combined compression‐vibration therapy using non‐invasive Beautylizer Therapy Cosmospheres V on the subcutaneous tissue morphology in women with gynoid lipodystrophy (pilot study). Journal of Cosmetic Dermatology, 1-6. https://doi.org/10.1111/jocd.14874

Mammucari, M., Russo, D., Maggiori, E., Paolucci, T., Di Marzo, R., Brauneis, S., ... & Natoli, S. (2021). Evidence based recommendations on mesotherapy: an update from the Italian society of Mesotherapy. La Clinica Terapeutica172(1), 37-45.

Sousa-De Sousa, D., Tebar Sanchez, C., Maté-Muñoz, J. L., Hernández-Lougedo, J., Barba, M., Lozano-Esteban, M. D. C., ... & García-Fernández, P. (2021). Application of Capacitive-Resistive Electric Transfer in Physiotherapeutic Clinical Practice and Sports. International Journal of Environmental Research and Public Health18(23), 12446.

Specific comments.

Title: It´s righ.

Abstract. Incorporate in the summary, a more precise sentence of the results.

Introduction. This section presents the problem in a coherent and clear manner with the correct support of the scientific literature. However, it is convenient to update the references, since there are different works related to the subject and no mention is made, and it would even be interesting to mention the different existing works related to technological advances in sport. Also, it could be a future study of review.

Methods. Modify the method section and incorporate the sections: Design.

  • Study design. To write the design section, we recommend that you take some of the following methodologists as references.

Ato, M., López-García, J. J., & Benavente, A. (2013). A classification system for research designs in psychology. Anales de Psicología/Annals of Psychology29(3), 1038-1059.

Montero, I., & León, O.G. (2007). A guide for naming research studies in psychology. International Journal of Clinical and Health Psychology, 7(3), 847-862.

Results. Summary of study data and table are correct.

Conclusion.  Differentiate the discussion of the main conclusions of the work. To do this, you must create this section. And modify the limitations of the study and locate them in said section at the end. Also, they must be direct, and highlight the main contributions of the study.

References. They should be reviewed and updated according to the publication standards.

Author Response

Dear reviewer,

We are grateful for the time and effort you took to write these review.

We have done our best to incorporate all of your suggestions and send back the revised manuscript. We believe the content is much improved and more clear for readers thanks to your comments.

In the following, please find our answers to your comments. 

We have introduced comments on the structure of the manuscript. the manuscript was submitted to the Journal of Clinical Medicine.

Thank you for the suggestion and the list of recommended works. We took into account additional work in football, which demonstrated the great popularity of this method of regeneration. Unfortunately, due to the already very extensive number of citations, MDPI does not recommend further extending them.

The work structure and study design were based on the scheme for systematic reviews provided by the MDPI publisher. Additionally, we applied the PRISMA guidelines, which are commonly used for this type of study.

We have revised individual sections of the manuscript following the suggestions for which we thank you very much

Yours sinserely,

Szczepan Wiecha on behalf of all authors.

Round 2

Reviewer 1 Report

Thank you for the opportunity to rate the article. In addition to the clarifications made. Authors may add limitations that not all search bases were consulted. For example, EMBASE.

Author Response

Dear reviewer,

Thank you very much, we add this information to the limitation. 

Yours sinserely,

Szczepan Wiecha on behalf of all authors